# Protein Profiling of Placental Extracellular Vesicles in Gestational Diabetes Mellitus

**DOI:** 10.3390/ijms25041947

**Published:** 2024-02-06

**Authors:** Neva Kandzija, Sophie Payne, William R. Cooke, Faheem Seedat, Roman Fischer, Manu Vatish

**Affiliations:** 1Nuffield Department of Women’s and Reproductive Health, University of Oxford, Oxford OX3 9DU, UK; neva.kandzija@wrh.ox.ac.uk (N.K.); sophie.payne@queens.ox.ac.uk (S.P.); william.cooke@wrh.ox.ac.uk (W.R.C.); faheem.seedat@wrh.ox.ac.uk (F.S.); 2Nuffield Department of Medicine, University of Oxford, OX3 7BN Oxford, UK; roman.fischer@ndm.ox.ac.uk

**Keywords:** syncitiotrophoblast extracellular vesicle, proteomics, gestational diabetes mellitus, placenta, pregnancy

## Abstract

Throughout pregnancy, some degree of insulin resistance is necessary to divert glucose towards the developing foetus. In gestational diabetes mellitus (GDM), insulin resistance is exacerbated in combination with insulin deficiency, causing new-onset maternal hyperglycaemia. The rapid reversal of insulin resistance following delivery strongly implicates the placenta in GDM pathogenesis. In this case–control study, we investigated the proteomic cargo of human syncytiotrophoblast-derived extracellular vesicles (STBEVs), which facilitate maternal–fetal signalling during pregnancy, in a UK-based cohort comprising patients with a gestational age of 38–40 weeks. Medium/large (m/l) and small (s) STBEVs were isolated from GDM (n = 4) and normal (n = 5) placentae using ex vivo dual-lobe perfusion and subjected to mass spectrometry. Bioinformatics were used to identify differentially carried proteins and mechanistic pathways. In m/lSTBEVs, 56 proteins were differently expressed while in sSTBEVs, no proteins reached statistical difference. Differences were also observed in the proteomic cargo between m/lSTBEVs and sSTBEVs, indicating that the two subtypes of STBEVs may have divergent modes of action and downstream effects. In silico functional enrichment analysis of differentially expressed proteins in m/lSTBEVs from GDM and normal pregnancy found positive regulation of cytoskeleton organisation as the most significantly enriched biological process. This work presents the first comparison of two populations of STBEVs’ protein cargos (m/l and sSTBEVs) from GDM and normal pregnancy isolated using placenta perfusion. Further investigation of differentially expressed proteins may contribute to an understanding of GDM pathogenesis and the development of novel diagnostic and therapeutic tools.

## 1. Introduction

Insulin signalling triggers glucose uptake by muscle and adipose tissue, maintaining normoglycaemia [1]. During the latter part of a healthy pregnancy, physiological insulin resistance develops, diverting glucose to the developing foetus, aiding growth and development [2,3]. There is a consequent increase in insulin production. Gestational diabetes mellitus (GDM) arises due to a combination of excessive insulin resistance and insulin deficiency [4]. Estimated to impact 13.9% of all pregnancies globally [5], GDM has short- and long-term deleterious consequences for mother and foetus. GDM increases the risk of perinatal complications such as macrosomia (with associated difficulties at delivery), stillbirth and postnatal hypoglycaemia [6,7]. Moreover, both mother and child are at a higher risk of developing metabolic syndrome in the long-term [8].

Notably, insulin resistance progressively rises with placental growth during the course of pregnancy but immediately following delivery of the fetoplacental unit, the insulin resistance characteristic of GDM is usually rapidly reversed [9]. This suggests that the placenta plays a central role in the pathogenesis of GDM. The placenta serves as an interface between the mother and the foetus and releases various factors to facilitate signalling to the maternal tissues, including the secretion of growth hormones contributing to the stimulation of downstream pro-inflammatory cytokines [10]. Although placental endocrine mediators were initially postulated to underlie GDM, Kirwan et al. found no meaningful association between placental hormones and insulin resistance [11]. This suggests the involvement of additional placental signalling molecules that may contribute to the pathophysiology of GDM.

Small and medium/large syncytiotrophoblast-derived extracellular vesicles (sSTBEVs and m/lSTBEVs) are released from the placenta into the maternal circulation during pregnancy [12]. These STBEVs carry genetic material and proteins to downstream tissues via circulation, facilitating communication between the placenta and the maternal tissue during pregnancy, which affects cell biology [13,14]. A recent study reports in vitro STBEV uptake by HEPG2 cells, with subsequent alterations in cholesterol biosynthetic gene expression and hepatocyte lipid metabolism [15]. Additionally, infusion of sSTBEVs obtained from women with GDM into nonpregnant mice decreased both insulin release from beta-islets and insulin sensitivity to a greater degree than infusion with sSTBEVs obtained from normoglycaemic controls [16]. Furthermore, insulin-stimulated glucose uptake and insulin-stimulated migration were attenuated when human skeletal myoblasts were co-cultured with sSTBEVs from GDM pregnancies compared to STBEVs derived from normal pregnancies [17].

STBEVs have already been shown to play a role in the pathogenesis of other pregnancy-associated diseases [18]. STBEVs are associated with the pro-inflammatory environment in pregnancy and m/lSTBEVs from preeclamptic patients induce an augmented inflammatory response in immune cells when compared to those obtained from healthy controls [19]. Hence, placental STBEV release was postulated to contribute to the development of GDM [20,21] and may represent a potential avenue for future therapeutics [22]. Further elucidation of the differences in composition and contents of STBEVs from women with GDM compared to STBEVs in normal pregnancy may thus prove vital in improving our understanding of the pathogenesis of GDM.

Our study aims to characterise and detect differences in the protein cargo of normal pregnancy and GDM STBEVs using mass spectrometry. Jayabalan et al. have previously investigated proteomic cargo in small extracellular vesicles (EVs) isolated from peripheral plasma samples of women undergoing OGTT. In total, 415 proteins were identified in their study, of which 78 were significantly and differentially expressed in sEV isolated from the plasma of women with GDM. The significantly different proteins included spectrin alphaerythrocytic (SPTA)-1, CAMK2β, PAPP-A, Perilipin 4, fatty acid-binding protein (FABP) 4 and hexokinase-3, all of which were associated with insulin resistance [23]. However, their study describes the protein cargo of all EVs isolated from peripheral plasma samples and not EVs specifically derived from the placenta. Hence, we focused on studying the differences of placenta-derived STBEVs at the proteomic level to potentially identify novel disease-causing or disease-reporting targets.

## 2. Results

### 2.1. Placental Lysates, sSTBEVs and m/lSTBEVs Can Be Characterised as Distinct Populations

To characterise STBEV populations, Western blots using primary antibodies specific to known extracellular vesicle markers were performed. Enhanced expression of CD63 and CD9 was seen in sSTBEVs compared to placental lysates (PLs) and m/lSTBEVs (Figure 1a), and Syntenin in sSTBEV samples (Figure 1b). The NDOG2 (in-house) antibody was also used to identify PLAP, indicating the placental origin of STBEVs, which was present in all STBEV samples.

To further characterise STBEV populations by size distribution and concentration profiles, NTA was performed. While there is some overlap between groups, a more prominent peak of molecules below 200 nm is seen in sSTBEV, with a mean size of 213.6 ± 2.8 nm, while particles in the m/lSTBEV sample are typically larger, with a mean of 256.0 ± 7.1 nm (Figure 1d,e). Together, these results are consistent with two separate populations of STBEVs, m/lSTBEVs and sSTBEVs.

### 2.2. Proteomic Characterisation of STBEVs from GDM and Normal Pregnancy

Proteomic analysis of STBEVs from normal and GDM pregnancies identified 1981 proteins in total. Filtering for valid values excluded seven proteins and 1974 proteins were used for the subsequent analysis. Prior to any statistical analysis, data were subjected to normalisation by median (Figure 2a). Hierarchical clustering was used to observe the correlation between the four groups. The strongest separation was observed between the two subtypes of EVs used in this study: m/lSTBEV and sSTBEVs in both GDM and normal pregnancies (Figure 2b). This was further supported by multi-scatter plot analysis presented with their corresponding Pearson correlation factor (Figure 2c). Finally, principal component analysis (PCA) was performed, confirming that EV proteins clustered based on their subtype (Figure 2d). We also observed that m/lSTBEV STBEVs from GDM and normal pregnancies clustered separately, while this was not the case for sSTBEVs. We then moved to establish the differences in the protein abundance between the different subtypes of EVs from GDM and normal pregnancies, respectively.

### 2.3. Differentially Expressed Proteins between m/lSTBEV and sSTBEVs

The comparative analysis between m/lSTBEV and sSTBEVs within GDM and normal pregnancies, respectively, demonstrated significantly different protein cargos between the two populations, suggesting that the biogenesis of these two types of EVs includes different cargo sorting mechanisms. In GDM, 856 proteins with unique peptides ≥ 2 were identified as being expressed differently (FDR-adjusted *p*-value). In total, 346 were up-regulated (log2 fold change ≥ 1), while 271 were down-regulated (log2 fold change ≤ −1). In normal pregnancies, 815 proteins with unique peptides ≥ 2 were differentially expressed between m/lSTBEV and sSTBEVs. Again, differentially expressed proteins were then further filtered by their log2 fold change (≤−1 or ≥1) to improve confidence in the data (Figure 3a). In normal pregnancy samples, 394 proteins were up-regulated in m/lSTBEV STBE-EVs, while 261 were down-regulated (Figure 3b).

### 2.4. Differentially Expressed Proteins in STBEVs from GDM and Normal Pregnancies

The comparative analysis of STBEV cargos between GDM and normal pregnancies was conducted for both m/lSTBEV and sSTBEVs, respectively. In m/lSTBEVs, the results showed that 56 proteins with unique peptides ≥ 2 were differently expressed (FDR-adjusted *p*-value) between GDM and normal pregnancies. Differentially expressed proteins were then further filtered by their log2 fold change (≤−1 or ≥1) to help increase the confidence in the data. Among them, six proteins were up-regulated (BANF1, CRYZL1, PSG3, ACPP, ANO10 and ITIH2) in GDM, while seven were down-regulated (TREML2, EIF3M, GBP1, RANBP1, PTGES3, CARHSP1 and OTUB1) (Figure 4a). Surprisingly, in sSTBEVs, no proteins were identified to be significantly different between GDM and normal pregnancies (Figure 4b). All differentially expressed proteins from m/lSTBEVs and their corresponding FDR-adjusted *p*-value (q-value) are listed in Table 1.

The functional annotation of differentially expressed proteins were assessed by g:Profiler analysis using Gene Ontology (GO) Metabolic Function (MF), GO Biological Processes (BP), GO Cell Components (CC) and Reactome datasets (Figure 5). GO:MF showed that 10 terms were significantly enriched with the top three terms being cadherin binding (GO:0045296), cell adhesion molecule binding (GO:0050839) and ubiquitin protein ligase binding (GO:0031625). In GO:BF, 11 terms were significantly enriched including. GO:CC reported 15 categories to be enriched, with the top three being cytosol (GO:005829), focal adhesion (GO:0005925) and cell–substrate junction (GO:0030055).

The REVIGO online tool was used to summarise and visualise the g:Profiler results. The data were presented on a scatter plot where semantically related GO terms are clustered together. Analysis of enriched terms from the GO:MF dataset revealed three major clusters: cadherin binding, ubiquitin protein ligase binding and cell adhesion molecule binding (Figure 6a). The same type of analysis was performed to visualise the g:Profiler data from GO:BF, identifying positive regulation of cytoskeleton organisation as a main cluster enriched in differentially expressed proteins between GDM and normal pregnancies in m/lSTBEVs (Figure 6b).

## 3. Discussion

In this study, we characterised protein profile changes in STBEVs isolated from GDM pregnancies in comparison to those isolated from normal pregnancies. We observed numerous differences in the protein cargo of m/lSTBEVs compared to sSTBEVs in both GDM (856 proteins) and normal pregnancies (815 proteins). Comparison analysis between GDM and normal pregnancies identified 56 proteins as differentially expressed in the m/lSTBEV subtype, while in the sSTBEV group, no proteins reached statistical significance. A challenge in STBEV research is the inability to isolate specific placental EVs. We are one of the few laboratories that perform ex vivo placenta perfusion, which allows for the isolation of placental-specific derived EVs. To the best of our knowledge, this work presents the first comparison of STBEVs’ protein cargo from GDM and normal pregnancies, isolated using placenta perfusion. Furthermore, we isolated two populations of m/lSTBEVs and sSTBEVs and described changes in the proteomic profile between them.

The three major subtypes of EVs are microvesicles (released by direct budding of the cell membrane in response to stimuli; previously considered size range: 200–1000 nm); exosomes (secreted by the fusion of MVB with cell membrane; size range: smaller than 200 nm); and apoptotic bodies (formed during the apoptosis; size range: above 1 µ). In recent years, it has become apparent that it is difficult to distinguish between these groups based on their size or using a conventional biomarker. Consistent with the latest International Society of Extracellular Vesicles (ISEV) guidelines, we describe two populations of STBEVs as m/l and sSTBEVs, separated based on size. Interestingly, our data illustrate that differences in protein profiles may provide a useful additional metric by which to clearly differentiate small from medium/large EV populations. The differences in protein cargo may be due to their different mode of biogenesis and it may be that altered functional effects that occur following downstream signalling by STBEVs are the result of different protein cargoes carried by distinct EVs. The functional enrichment analysis performed by g:Profiler [24] was filtered for redundant terms and summarised using REVIGO software [25]. Enriched terms in GO biological processes showed that differentially expressed proteins between m/lSTBEV and sSTBEVs were associated with the immune system and leukocyte-mediated immunity in both GDM and normal pregnancies. This is further supported by the Reactome database analysis [26], where the most significantly enriched term was associated with the innate immune system again in both GDM and normal pregnancies. This finding is in accordance with previous research that has described placentally derived microvesicles (m/lSTBEVs) as immunostimulatory particles while exosomes (sSTBEVs) exhibit immunosuppressive properties [27]. It is postulated that this might be important in complicated pregnancies as discussed in more detail below.

When comparing the proteomic cargo of the same subtype of STBEVs between GDM and normal pregnancies, amongst m/lSTBEVs, 56 proteins were statistically different and of these, 13 had a log2 fold change ≤ −1 or ≥1. For sSTBEVs, however, there were no statistically significant differentially expressed proteins between GDM and normal pregnancies. These data support the hypothesis that sSTBEVs (previously classified as exosomes) that are continuously released from the placenta are predominantly expressed in normal pregnancy, while m/lSTBEVs (previously described as microvesicles) are produced as a result of inflammatory and oxidative stress in the diseased placenta that stimulates the STB to release m/lSTBEVs [27]. In preeclampsia, it was suggested that inflammatory and oxidative stress prompt the shedding of microvesicles, subsequently shifting the balance towards a greater number of larger m/lSTBEVs, compared to sSTBEVs; the most prominent subgroup in normal pregnancy [27].

Functional enrichment analysis of differentially expressed proteins in m/lSTBEVs from GDM and normal pregnancies notes a positive regulation of cytoskeleton organisation as the most significantly enriched biological process. Enriched cytoskeleton organisation was also reported in other EVs confirming that vesicular proteins mainly originate from the plasma membrane, cytosol and internal vesicles, and as such are distinct from other intracellular compartments (i.e., endoplasmic reticulum, mitochondria, etc.) [28].

Studying STBEV cargo has multiple benefits. It provides insights into the role of STBEVs in disease pathophysiology; moreover, changes to EV protein cargo in disease could provide novel biomarkers to aid in more reliable diagnosis. This is particularly relevant in GDM due to the limitations of current diagnostic tools. For example, we postulate that unique proteins observed amongst GDM-specific STBEVs isolated by placental perfusion may also be detected in the plasma due to the increased placental sSTBEV quantities seen in GDM pregnancy [29].

Among the 13 proteins that had adjusted *p*-value < 0.05 and log2 fold changes ≤ −1 or ≥1, some warrant special interest due to their expression patterns and associations with processes relevant to the pathophysiology of GDM. A Human Protein Atlas search for tissue expression [30] showed that PSG-3 is exclusively expressed by the placenta and may have potential as a biomarker. Higher PSG-3 levels are reported in preeclampsia, another gestational disease associated with increased insulin resistance [31]. Similarly, GBP1 expression appears to increase with progressing healthy pregnancies, but this increase is attenuated in preeclampsia, associating reduced GBP1 expression, as reported by proteomics in this study, with an inflammatory placental environment [32].

BANF-1/BAF-1 is a nuclear reassembly factor [33,34], capable of dampening cGAS-STING signalling by reducing cytosolic levels of double-stranded DNA in microglial cells [35]. This pro-inflammatory pathway is activated in placental mitochondria during GDM [36], potentially augmenting placental stress. It is thus possible that BANF-1 up-regulation occurs as a protective mechanism in GDM, reducing cGAS-STING signalling to prevent further placental stress and representing a possible target for novel therapeutics.

In addition to inflammation, some of the proteins identified in this study appear to interact with metabolic pathways. P23 was implicated in the metabolism of lipids, fatty acids and prostaglandins [37,38], and may play an ambiguous role in glucose metabolism [39]. CARHSP1 expression, rather, may be associated with both glucose metabolism [40] and inflammatory signalling [41]. SIRNA knockdown of CARHSP1 in hepatocytes increases gluconeogenic gene expression [42]. The gluconeogenic capacity of the placenta is debated [43,44]; however, down-regulation of CARHSP1 in the placenta, indicated by the apparent down-regulation in STBEVs, or globally in GDM could increase gluconeogenesis and thus contribute to the hyperglycaemic characteristic of GDM.

Limitations: this discovery study is limited by small numbers but placenta perfusion is a technically complex procedure that has a 1:4 chance of succeeding. In addition, patient selection was specifically confined to GDM patients on medication who were undergoing an elective cesarean section and who complied with our inclusion/exclusion criteria.

We obtained STBEV samples using dual-lobe placental perfusion from patients at term. EVs derived from the placenta at different points during pregnancy may well have different profiles [19,45]. Clearly, it is not ethical to acquire pre-term GDM placentas for research but data from our study can be used to interrogate plasma samples from earlier in pregnancy, which is ethically sound.

## 4. Materials and Methods

### 4.1. Human Subjects

This project was approved by the Central Oxfordshire Research Ethics Committee C (REFS 07/H0607/74 and 07/H0606/148). Participants provided written informed consent and full-term [46] placentas were collected and perfused following elective caesarean section. Inclusion criteria for this study was GDM diagnosed by plasma glucose measuring levels ≥ 7.1 mmol/L during fasting, and/or plasma glucose of ≥10.1 mmol/L at 1 h and/or ≥8.5 mmol/L at 2 h following an oral glucose load of 75 g (GDM guidelines for the Oxford University Hospital). The control group included gestational age and birth weight-matched pregnancies without abnormal oral glucose tolerance test results (Table 2). Patients with pre-pregnancy diabetes, cardiovascular disease or preeclampsia were excluded from this study.

### 4.2. Isolation and Characterisation of STBEVs

Control and GDM STBEVs were isolated using serial ultracentrifugation following dual-lobe placental perfusion [47], modified according to a previously defined protocol [48]. M/lSTBEV STBEVs were obtained from maternal placental perfusate centrifuged at 10,000× g, while pellets obtained by 150,000× g centrifugation yielded samples enriched in sSTBEVs. STBEVs were characterised using Nanoparticle Tracking Analysis, bicinchoninic acid assay (BCA) and Western blotting as described below. Following BCA protein assay to determine concentration, samples were stored at −80 °C, before being moved to −20 °C storage for the duration of the study.

### 4.3. Nano-Particle Tracking Analysis (NTA)

NTA was performed as previously described [49]. To characterise STBEV populations, samples were diluted with filtered phosphate-buffered saline (fPBS) then measured using a NanoSight NS500 (Malvern Instruments, Malvern, UK) using an inbuilt sCMOS camera and NTA software version 2.3, build 0033 (Malvern Instruments, Malvern, UK). Prior to sample loading, silica 100 nm microspheres (Polysciences, Warrington, UK) were passed through the instrument for calibration. A camera level of 12 was used, with a 405 nm laser and 5 recordings of 30 s were analysed to give the approximate diameter and concentration of sSTBEVs and m/lSTBEVs.

### 4.4. Bicinchoninic acid Protein Assay

BCA protein assays were performed according to an early protocol defined by Brown et al. [50] to calculate sample protein concentrations. Samples were diluted with fPBS (1/10 for PL, ½ for STBEVs) and added to a 96-well plate alongside standard solutions (0, 0.0625, 0.125, 0.25, 0.5, 1 and 2 mg/mL). Sample layout was recorded and BCA reagent (50 BCA A: 1 BCA B) (Pierce, ThermoScientific, Waltham, MA, USA) was added before mixing and incubation (30 m at 37 °C). The plate was cooled to r/t and colour intensity was measured at 562 nm (BMG Labtech FLUOstar OPTIMA, BMG LabTech, Aylesbury, UK), using Optima Data Analysis to produce a protein standard curve, from which sample protein concentration was calculated.

### 4.5. Mass Spectrometry

Mass spectrometry analysis of STBEVs obtained from placenta perfusion was performed in collaboration with the Target Discovery Institute (TDI), University of Oxford, as previously described [51]. After isolation and characterisation of STBEVs, both m/lSTBEV STBEVs and sSTBEVs from GDM (n = 4 for both STBEV subtypes) and normal (n = 5 for both STBEV subtypes) pregnancy were submitted to TDI.

Samples were thawed on ice and lysed in RIPA buffer for 30 min (also on ice) followed by centrifugation at maximum speed. Clear supernatants were collected for each sample. Samples were then reduced with 5 mM dithiothreitol for 30 min, followed by alkylation with 20 mM iodoacetamide for another 30 min. Then, methanol–chloroform was used for precipitation. Pellets were resuspended in 1 M (final concentration) urea and digested with approximately 0.85 g trypsin at 37 °C overnight. Digest samples were acidified to 1% formic acid and desalted on Sola Cartridges. Collected eluates were dried to completeness in a SpeedVac.

Samples were resuspended in 2% acetonitrile/0.1% formic acid and diluted 1:20 prior to injection. Peptides were injected into a liquid chromatography–mass spectrometry system (LC–MS) comprised of a Dionex Ultimate 3000 Nano LC and a Thermo Q-Exactive mass spectrometer. Peptides were separated on a 50 cm long EasySpray column (ES803; Thermo Fisher, Milton Keynes, UK) with a 75 µm inner diameter and 60 min gradient of 2% to 35% acetonitrile in 0.1% formic acid and 5% dimethyl sulfoxide (DMSO) at a flow rate of 250 nL/min.

MS1 spectra were acquired with a resolution of 70,000. The top 15 most abundant peaks were fragmented after isolation with a mass window of 1.6 Th at a resolution of 17,500.

### 4.6. Statistics

Data analysis was carried out using Progenesis (Nonlinear/Waters) for quantification and Mascot for protein identification (against UPR_HomoSapiens). Perseus (https://maxquant.net/perseus/ accessed on 2 March 2020) was used for subsequent analysis (Tyanova and Cox, 2018). Data were log2 transformed, filtered for valid values (70% valid values in at least one group) and normalised by median. Imputation was used to replace missing values from a normal distribution. Comparison analyses between the two groups were carried out with two sample *t*-tests with permutation-based false discovery rate (FDR) correction *p* < 0.05. Perseus online software (1.6.10.50).was also used to generate principal-component analysis (PCA) using singular value decomposition as described in Tyanova et al. [52] and Pearson correlation was calculated to assess the strength of the linear relationships between different STBEVs populations.

Functional enrichment analysis was carried out using g:Profiler (https://biit.cs.ut.ee/gprofiler accessed on 2 March 2020) online software [24] on the differentially expressed proteins from STBEVs (*p* < 0.05). g:Profiler analyses uses cumulative hypergeometric testing with a g:SCS (Set Counts and Sizes) correction method.

Next, GO-enriched terms generated in previous step were further filtered for redundant terms using REVIGO’s default settings (https://revigo.irb.hr/ accessed on 2 March 2020) [25] to obtain a summary of GO terms between m/lSTBEVS from GDM and normal pregnancies.

### 4.7. Western Blotting

Western blotting was performed according to a previously defined protocol [51]. STBEV and placental lysate (PL) samples were appropriately diluted according to BCA results to a concentration yielding a constant total protein mass (30 µg placental lysate blots, 20 µg characterisation blots and STBEVs) and heated with Laemmli reducing or non-reducing buffer (Bio-Rad, Hercules, CA, USA). Prepared samples were loaded on a 10% SDS-PAGE gel, with the first well loaded with Precision Plus ProteinTM ladder (Bio-Rad) and run at 150 V for 1 h 15 m, before transfer to a polyvinylidene fluoride (PVDF) (ThermoScientific) membrane at 25 V for 45 m. Membranes were stained with Ponceau Red-S (ThermoScientific) and a scanned .tif image of this stage was used to control for protein loading in immunoblotting analysis. Following this, membranes were blocked with 5% milk and incubated overnight at 4 °C with primary antibodies (Appendix A). All antibodies were validated by the manufacturer (respective data sheets). Membranes were washed with tris-buffered saline (TBS) and Tween20 (TBS-T) and incubated for 1 h at r/t with species-specific HRP-labelled secondary antibodies (Appendix A). Bands were subsequently visualised with enhanced chemiluminescence detection reagents PLUS (Pierce, Thermo Scientific) using G:BOX (Syngene, Synoptics Limited, Cambridge UK).

## 5. Conclusions

We present the first report of direct placental perfusion comparison of m/lSTBEVs and sSTBEVs between normal and GDM patients. This has yielded a number of proteins that are significantly different between normal and diseased types. It is likely that these are either a potential driver of GDM or are expressed as a result of the disease. Further studies are necessary to validate these results in a number of different techniques, as well as a larger cohort, and investigate the function and implications of the differentially expressed proteins in GDM pregnancies. Understanding whether these differences are present earlier in gestation may have biomarker potential given the relatively poor performance of the oral glucose tolerance test. STBEVs have great potential to provide a non-invasive, real-time snapshot of placenta health, providing both novel diagnostic and treatment opportunities for gestational diseases and enhancing our understanding of the mechanism involved in GDM’s pathophysiology.

## Figures and Tables

**Figure 1 ijms-25-01947-f001:**
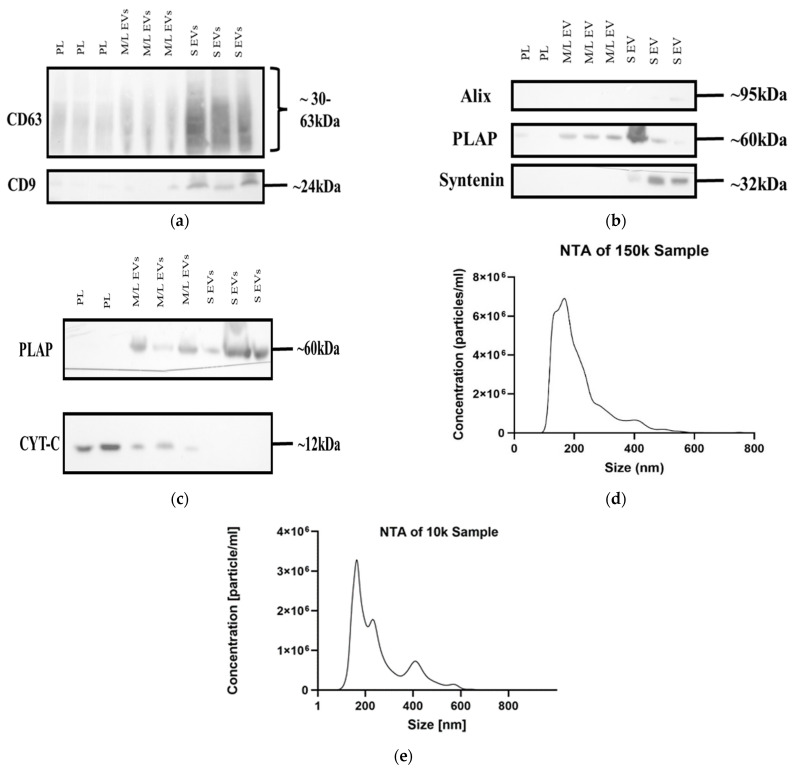
STBEV characterization: (**a**) Western blot of CD63 and CD9 expression in samples of Placental Lysate (3 × PL), m/lSTBEVs (3 × M/L EVs) and sSTBEVs (3 × S EVs); (**b**) Western blot analysis of Alix, PLAP and Syntenin expression in samples of Placental Lysate (2 × PL), m/lSTBEVs (3 × M/L EVs) and sSTBEVs (3 × S EVs); (**c**) PLAP and Cyt C expression in samples of Placental Lysate (2 × PL), m/lSTBEVs (3 × M/L EVs) and sSTBEVs (3 × S EVs). Full blots and ponceau red stains (Appendix A); (**d**) NTA particle size distribution (150k sample—sSTBEVs); (**e**) NTA particle size distribution (10k sample—m/lSTBEVs).

**Figure 2 ijms-25-01947-f002:**
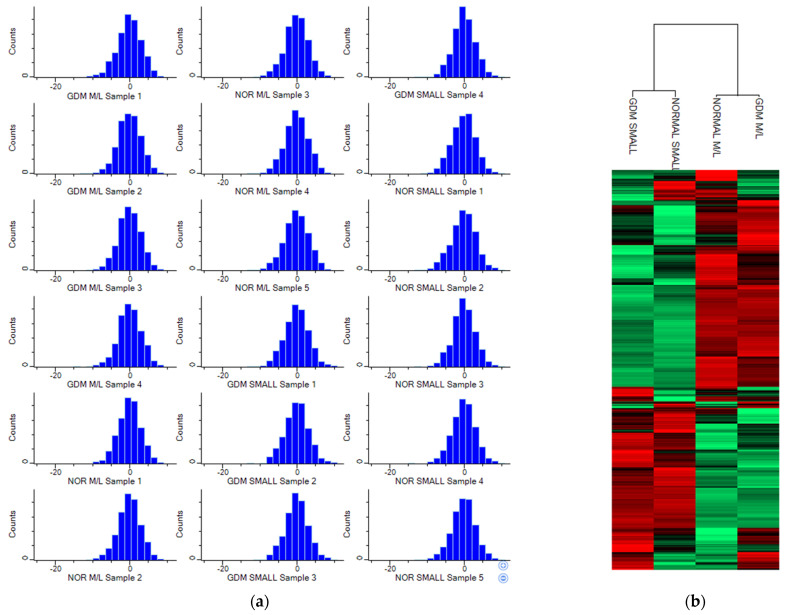
(**a**) Histograms of normalised protein abundance for all samples; (**b**) hierarchical clustering of m/lSTBEVs and sSTBEVs from GDM and normal pregnancy; (**c**) multi-scatter plot of m/lSTBEVs and sSTBEVs isolated from GDM and normal pregnancies with the corresponding Pearson correlation factor (R). The strongest differences are observed between m/lSTBEVs and sSTBEVs from GDM (R = 0.835) and m/lSTBEVs and sSTBEVs from normal pregnancies (R = 0.837); (**d**) principal component analysis (PCA) for all samples. PCA showed the largest variance between two subtypes of STBEVs. m/lSTBEVs from both GDM (blue dots) and normal pregnancies (pink dots) clustered separately from sSTBEVs from GDM (green dots) and normal pregnancies (purple dots).

**Figure 3 ijms-25-01947-f003:**
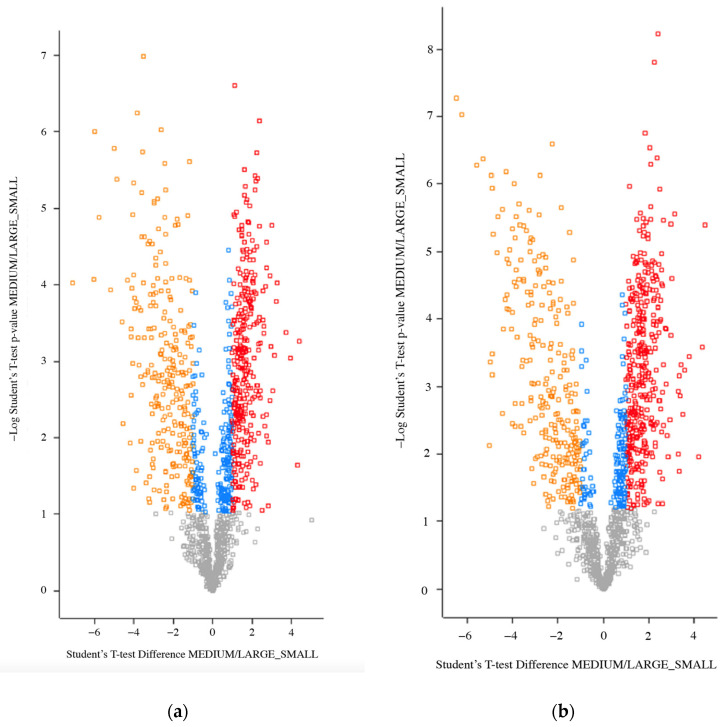
Comparison of protein profile between m/lSTBEVs and sSTBEVs. The above Volcano plots show differentially expressed proteins for m/lSTBEVs and sSTBEVs isolated from GDM (**a**) and normal pregnancies (**b**). Protein abundance was analysed with Student’s *t*-test and differentially expressed proteins (FDR-adjusted *p*-value < 0.05) with log2 fold change ≤ −1 or ≥1 are presented in yellow (down-regulated) and red (up-regulated). Differentially expressed proteins without log2 fold change ≤ −1 or ≥1 are presented in blue. Differentially expressed proteins that did not reach statistical significance are presented by grey squares. Images were produced using Perseus online tool (1.6.10.50).

**Figure 4 ijms-25-01947-f004:**
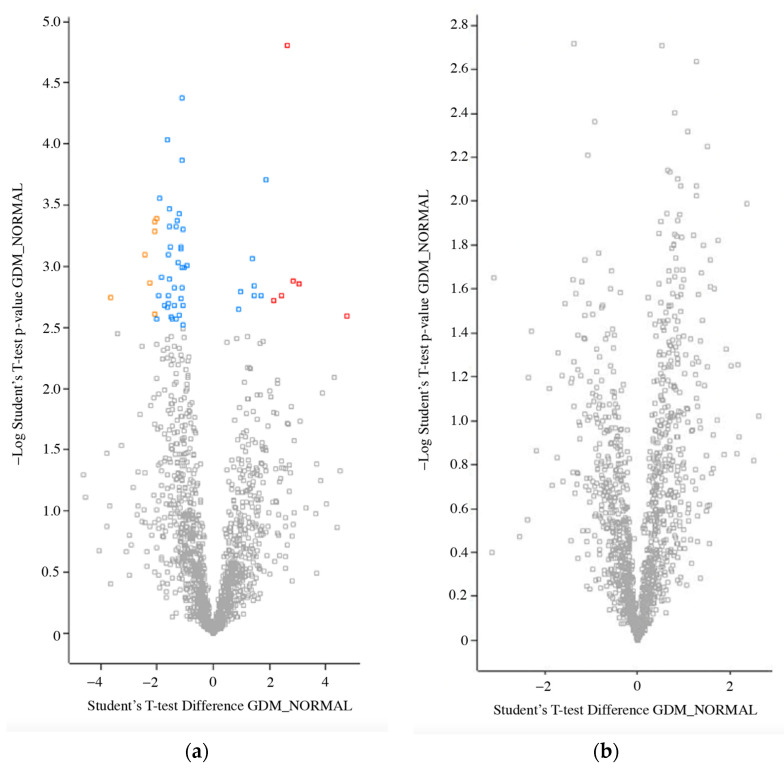
Comparison of STBEV protein profile between GDM and normal pregnancies in m/lSTBEVs (**a**) and in sSTBEVs (**b**). Protein abundance was analysed with Student’s *t*-test and differentially expressed proteins (FDR-adjusted *p*-value < 0.05) with log2 fold change ≤ −1 or ≥1 are presented in yellow (down-regulated) and red (up-regulated). Differentially expressed proteins (FDR-adjusted *p*-value < 0.05) without log2 fold change ≤ −1 or ≥1 are presented in blue. Differentially expressed proteins that did not reach statistical significance are presented by grey squares. Images were produced using Perseus online tool.

**Figure 5 ijms-25-01947-f005:**
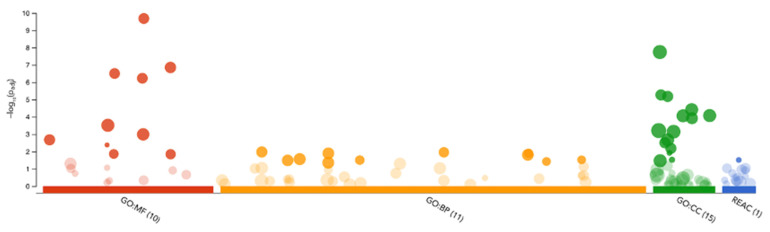
Manhattan plot of functional enrichment analysis of differently expressed proteins in GDM m/lSTBEVs. Differentially expressed proteins were subjected to g:Profiler analysis. The software based its analysis on Gene Ontology (GO), Metabolic Function (MF), GO Biological Processes (BP), GO Cell Components (CC) and Reactome datasets, colour coding each individual dataset for increased clarity. Each dot on the Manhattan plot above represents a single functional term. The size of the dot is scaled according to the number of annotated genes in that term. The darker the colouring, the more significant the terms. Analysis of the GO:MF dataset found that 10 terms were significantly enriched; in the GO:BP dataset, this was 11 terms; and in the GO:CC, 15 terms (indicated by the dark red, yellow and green dots, respectively). A single term was enriched significantly in the Reactome dataset, shown here in blue.

**Figure 6 ijms-25-01947-f006:**
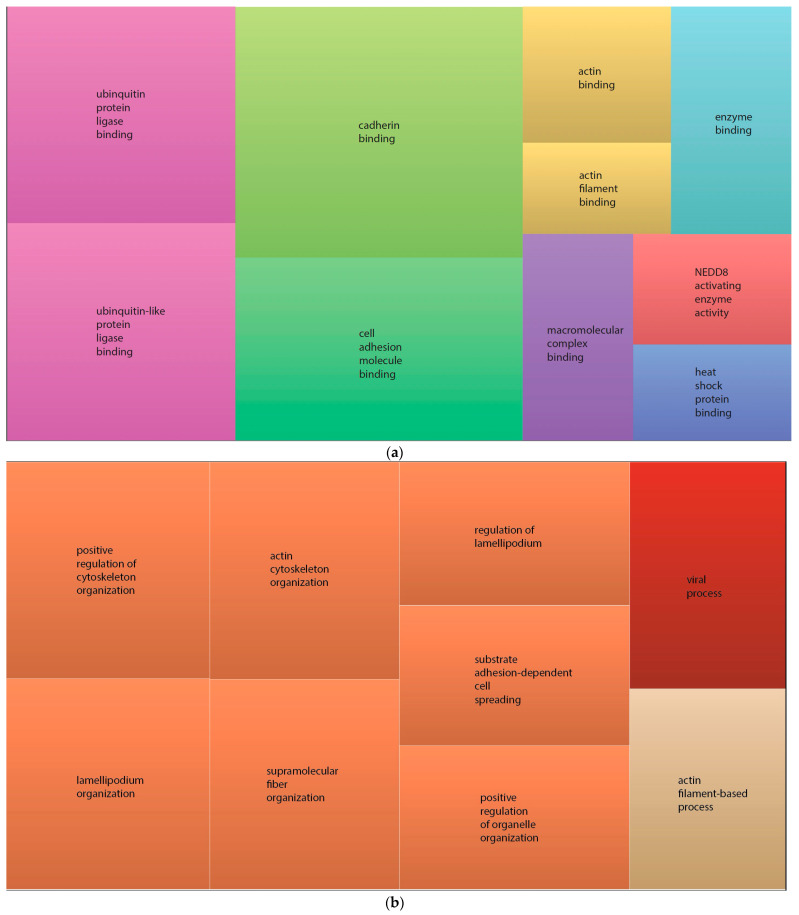
Summary of GO-enriched terms between m/lSTBEVS from GDM and normal pregnancies. GO-enriched terms obtained by g:Profiler analysis of differentially expressed proteins between GDM and normal pregnancies in m/lSTBEVS were summarised and visualised using REVIGO. Summary of GO-enriched terms in MF (**a**) and BP (**b**). Every rectangle represents a single cluster. The representatives are also linked to a ‘superclusters’ of roughly related terms indicated by the same colour. Size of the rectangles reflects the frequency of the GO term in the GOA database (larger rectangles are associated with more general terms).

**Table 1 ijms-25-01947-t001:** Differentially expressed proteins (FDR-adjusted *p*-value < 0.05) between MEDIUM/LARGE STBEVs from GDM and normal pregnancies. Proteins with log2 fold change ≤ −1 or ≥1 are presented in blue (down-regulated) and red (up-regulated).

Protein	q-Value	*p*-Value	Difference
** Barrier-to-autointegration factor **	0.008	1.5867 × 10^−5^	1.30996013
**COP9 signalosome complex subunit 4**	0.024	4.1937 × 10^−5^	−0.5596797
** Calcium-regulated heat-stable protein 1 (Fragment) **	0.02586667	0.00051904	−1.0356117
**GMP reductase**	0.02707692	0.00047596	0.6525968
**Cullin-5**	0.02714286	0.00050147	−0.5409105
**Actin-related protein 2/3 complex subunit 3**	0.02933333	0.00047484	−0.7771788
**Eukaryotic translation initiation factor 3 subunit M (Fragment)**	0.03018182	0.00042916	−1.0389393
** Guanylate-binding protein 1 **	0.0314	0.00080078	−1.2220971
**UV excision repair protein RAD23 homolog B**	0.03221053	0.00078986	−0.7934964
**Aldose reductase**	0.03223529	0.00069685	−0.5726541
**Isoform 3 of Twinfilin-1**	0.03288889	0.00071103	−0.5767815
**Isoform 3 of Reticulon-4**	0.0332	0.00042166	−0.6465006
**UMP-CMP kinase**	0.03425	0.00068068	−0.7637424
**Integrin beta-1**	0.03657143	0.00086327	0.68686013
** Prostaglandin E synthase 3 **	0.03688889	0.00040225	−1.0072526
**Elongin-B**	0.03756522	0.00096094	−0.4621809
**T-complex protein 1 subunit delta**	0.0376	0.00019572	0.93678346
**Isoform 2 of Elongation factor 1-gamma**	0.03776	0.0010248	−0.5571422
**Alpha-parvin**	0.038	0.00101189	−0.5276663
**Isoform 2 of Kinesin light chain 2**	0.03836364	0.00091909	−0.6181082
**Cell division control protein 42 homolog**	0.039	0.00013928	−0.560149
**Tubulin-specific chaperone A**	0.03991489	0.00213496	−0.8182631
**Isoform 1 of Deaminated glutathione amidase**	0.04017391	0.00209093	−0.8653197
**Sorting nexin-9**	0.0402	0.00177909	0.72804856
**Eukaryotic translation initiation factor 2 subunit 3**	0.04033333	0.00220814	0.45025353
**NEDD8-activating enzyme E1 catalytic subunit**	0.04063158	0.00174762	−0.9790669
** Trem-like transcript 2 protein **	0.04102564	0.00177836	−1.8172884
** Quinone oxidoreductase-like protein 1 **	0.04104762	0.00191464	1.07774053
**Activator of 90 kDa heat shock protein ATPase homolog 1**	0.04106667	0.0020824	−0.6881804
**14-3-3 protein epsilon**	0.04126829	0.0018585	−0.5699852
**Isoform 1B of Beta-arrestin-1**	0.0415	0.00036855	−0.6159174
**Serine/threonine-protein phosphatase CPPED1**	0.04152727	0.00269118	−0.7273863
** Pregnancy-specific beta-1-glycoprotein 3 **	0.04172973	0.00174198	1.2087852
**Actin-related protein 2/3 complex subunit 2**	0.042	0.00207294	−0.5355681
** Isoform 2 of Prostatic acid phosphatase **	0.04223077	0.00255752	2.38245492
**Isoform 2 of Coronin-1C**	0.0422963	0.00268045	−0.6556026
**Cullin-2**	0.04242424	0.0014969	−0.6892327
**Immunity-related GTPase family Q protein**	0.04266667	9.0978 × 10^−5^	−0.8085251
**Keratin, type I cytoskeletal 18**	0.04266667	0.00172857	0.84190021
**Trophoblast glycoprotein**	0.0427907	0.0020254	−0.8009386
**Spartin**	0.04279245	0.00265268	−0.9998695
**Monocarboxylate transporter 4**	0.042875	0.0014769	−0.5568041
**Isoform 2 of Ras-related protein Rab-4B**	0.04290196	0.00254245	−0.7388124
**Isoform 2 of Receptor-type tyrosine-protein phosphatase alpha**	0.04305882	0.0016122	0.48871684
**Isoform 2 of Destrin**	0.04333333	0.00027368	−0.9514709
**Isoform 2 of Serine/threonine-protein phosphatase 2A 56 kDa regulatory subunit epsilon isoform**	0.04352	0.00248797	−0.6112685
**Myristoylated alanine-rich C-kinase substrate**	0.04388571	0.00170482	−0.8004272
**Vesicular integral-membrane protein VIP36**	0.04425806	0.00145655	0.73318152
** Ran-specific GTPase-activating protein **	0.04440816	0.00247769	−1.0355191
** Isoform 2 of Anoctamin-10 **	0.0448	0.0013966	1.51864133
** Ubiquitin thioesterase **	0.04482759	0.00136681	−1.1312524
**Pyruvate kinase PKM**	0.04514286	0.00033608	−0.7798404
**Exportin-7**	0.04533333	0.00127124	−0.7891689
**Isoform 2 of NEDD8-activating enzyme E1 regulatory subunit**	0.04584615	0.00121286	−0.9162508
** Inter-alpha-trypsin inhibitor heavy chain H2 **	0.04642857	0.001324	1.41273007
**Heat shock 70 kDa protein 1A**	0.04807143	0.00300704	−0.5472665

**Table 2 ijms-25-01947-t002:** Clinical data of m/lSTBEV and sSTBEV samples used for mass spectrometry (discovery cohort). Data presented as Mean ± SEM, (CI).

	GDM (n = 4)	Control Pregnancy (n = 5)
Age, year	34.8 ± 3.9	31 ± 1.2
	(24–42)	(28–35)
Gestation age, weeks	38.6 ± 1.0	39.1 ± 0.3
	(38.1–39.1)	(39.0–39.3)
Body mass index, kg/m^2^	27.7 ± 1.5	27.6 ± 1.4
	(25–32)	(25–32)
Newborn weight, g	3336.3 ± 386.2	3379.0 ± 175.9
	(2840–4450)	(2865–3890)
Max. systolic pressure, mmHg	132.8 ± 6.1	125.8 ± 4.3
	(122–150)	(110–135)
Max. diastolic pressure, mmHg	77.3 ± 5.7	82.4 ± 1.2
	(61–86)	(80–86)
GDM treatment	Metf: 2	N/A
	Ins + Metf: 2	

## Data Availability

M.V. is the guarantor of this work and, as such, has full access to all the data in the study and takes responsibility for the integrity of the data and the accuracy of the data analysis. All data analysed during this study are included in the published article and its online Appendix A. Proteomics results are available at https://figshare.com/articles/dataset/Proteins_GDM_EVs_xlsx/24922080 (accessed on 25 January 2024).

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
