# Peer review of "Protein Profiling of Placental Extracellular Vesicles in Gestational Diabetes Mellitus"

_ijms, 2024, doi:10.3390/ijms25041947_

Round 1

Reviewer 1 Report

Comments and Suggestions for Authors

This is a comprehensive study looking at the differentially expressed proteins between STBEVs of GDM placenta and normal pregnant placenta of women.

However, I have some concerns, which need to be justified

1. Only 4 GDM patients are included for the study, which seems less.

2. The discussion part lacks the discussion on differentially expressed proteins and their relevance to GDM pathogenesis.

3. Besides western blot, one more validation technique could be followed to get conclusive results.

Justify these points  

Author Response

Thank you very much for your helpful comments. Our responses are written in blue.

This is a comprehensive study looking at the differentially expressed proteins between STBEVs of GDM placenta and normal pregnant placenta of women. 

However, I have some concerns, which need to be justified 

1. Only 4 GDM patients are included for the study, which seems less. 

We understand that at first glance this may seem low, but placental perfusion is technically a complex procedure and only 1 in 4 placentae are successfully perfused. Additionally, we have reported results that are statistically significant (the minimum number for this is n=3). This is very different to obtaining plasma or serum samples which require less effort, but which are less discriminatory since the vesicle origin is less easy to infer. The unique advantage of placental perfusion is that the vesicles are (by definition) placental in origin.  

Furthermore, recruitment of women with GDM, who are then on treatment and who then have an elective cesarean section makes this even more challenging. This is why this (to the best of our knowledge) is the first report of placental vesicle comparison between normal and GDM using this technique. 

We have, however, now included this as a limitation of our study.  

  1. The discussion part lacks the discussion on differentially expressed proteins and their relevance to GDM pathogenesis. - SOPHIE

In light of this comment we have added some discussion on how some of the proteins identified could be relevant to the pathogenesis of GDM. We have focused here on significantly upregulated or downregulated proteins, which have links to inflammatory or metabolic cascades. 

  1. Besides western blot, one more validation technique could be followed to get conclusive results.

We agree with the reviewer but feel that this is beyond the scope of this current “discovery” research. As we move forwards we fully intend to characterize these proteins in a number of different techniques in a larger cohort. 

We fully take on board your comments and have added a sentence in the conclusion reflecting this. 

Reviewer 2 Report

Comments and Suggestions for Authors

Dear Authors,

Very modern and useful manuscript, thanks for it. I have just some minor comments:

1) Material and methods. Please, give the reference for the gestation week classification. If you have used the classification according to World Health Organization what is as follows: full-term infants – GA between 37 weeks and 41 week and 6 days, please, give the reference for it!

2) I would like to invite you to give references also for the all methods used in the manuscript, - for subsections 4.2, 4.3, 4.4., 4.5, 4.6. More likely your manuscript will be cited quite often and these references have to be placed there!

3) Add, please, the Limitation section at the end of Discussion and separate Conclusions from the Discussion (the future detection of STBEVs in different preterm placenta could be one of the limitation, by the way...)

4) References. Well, not very many, what is understandable in this relatively new field, but the M+M references will expand this part, too.

Otherwise, thank you and good luck with your future research!

Author Response

Dear Authors, 

Very modern and useful manuscript, thanks for it. I have just some minor comments: 

Thank you very much for your helpful comments. Our responses are written in blue.

1) Material and methods. Please, give the reference for the gestation week classification. If you have used the classification according to World Health Organization what is as follows: full-term infants – GA between 37 weeks and 41 week and 6 days, please, give the reference for it! - Neva 

Thank you for this observation, we used the WHO classification of term pregnancy (37-41+^) weeks and have referenced this in the text (line 353). 

2) I would like to invite you to give references also for the all methods used in the manuscript, - for subsections 4.2, 4.3, 4.4., 4.5, 4.6. More likely your manuscript will be cited quite often and these references have to be placed there!  

We agree with the reviewer on this matter and have now added references for the methods accordingly. Thank you for bringing this to our attention. 

3) Add, please, the Limitation section at the end of Discussion and separate Conclusions from the Discussion (the future detection of STBEVs in different preterm placenta could be one of the limitation, by the way...) -  

Thank you for this comment, we have now added a distinct limitations section and separated the conclusions from the discussion as you suggested. 

4) References. Well, not very many, what is understandable in this relatively new field, but the M+M references will expand this part, too. 

Thank you, we have now addressed this. 

Otherwise, thank you and good luck with your future research! 

Many thanks!

Reviewer 3 Report

Comments and Suggestions for Authors

·         Abstract: Suggest include the following (1) the country/place where the study was conducted; (2) the date of sampling took place; (3) the age of participants; (4) the type of study design (case-control); and (5) the statistical analysis used. The conclusion does not propose a clear direction for future studies (Line 22-23).

·         Lines 28-33; Lines 47-51: Need references here.

·         Lines 38-46: It would be benefit to focus on placental GDM as the major secretion site of growth hormones that play a role in stimulating pro-inflammatory cytokines. Please refer to this article (Nutrients. 2022 Dec 23;15(1):70).

·         Line 51: Recent studies? Study?

·   Lines 51-52: This in vitro study contains little synthesis of the information.

·         Line 59-63: I would suggest referring to this article (Front Physiol. 2019 Oct 1:10:1236).

·     There are missing references to literature, which are closely related to this work (Placenta. 2021 Sep 15; 113: 15-22; Diab Vasc Dis Res. 2022 Mar-Apr; 19(2): 14791641221093901; Front Cell Dev Biol. 2022; 10: 1060850; Clin Sci (Lond). 2023 Aug; 137(16): 1311–1332).

·         Authors should clarify the statistical tests used in their data analysis and the rationale for using those tests. For example, need to expand on PCA. Please describe further for the Kaiser criterion and the varimax method. Why Pearson correlation analysis/factor was necessary? The statistical analysis should be in a separate section (Lines 356-368).

·         Figure 2 C should be clear enough to the reader.

·         Lines 300-301: How exactly were the participants recruited? What were the inclusion/exclusion criteria?

·         Lines 274-282: Authors acknowledge some limitations of the study, but these are unclear and should be further discussed.

·         Lines 287-292: The study comes to a weak conclusion. The conclusions would benefit from giving consideration to how the paragraphs are structured and the thesis of each paragraph.

Author Response

Thank you very much for your helpful comments. Our responses are written in blue. 

  • Abstract: Suggest include the following (1) the country/place where the study was conducted; (2) the date of sampling took place; (3) the age of participants; (4) the type of study design (case-control); and (5) the statistical analysis used. The conclusion does not propose a clear direction for future studies (Line 22-23).

Thank you for these suggestions, we have now updated the abstract in light of these comments and hope that this will give a more conclusive overview the paper. 

  • Lines 28-33; Lines 47-51: Need references here. We have added this to the text.
  • Lines 38-46: It would be benefit to focus on placental GDM as the major secretion site of growth hormones that play a role in stimulating pro-inflammatory cytokines. Please refer to this article (Nutrients. 2022 Dec 23;15(1):70). We have added this to the text.
  • Line 51: Recent studies? Study? We have added this to the text.  
  • Lines 51-52: This in vitro study contains little synthesis of the information. We have added this to the text.
  • Line 59-63: I would suggest referring to this article (Front Physiol. 2019 Oct 1:10:1236). We have added this to the text.
  • There are missing references to literature, which are closely related to this work (Placenta. 2021 Sep 15; 113: 15-22; Diab Vasc Dis Res. 2022 Mar-Apr; 19(2): 14791641221093901; Front Cell Dev Biol. 2022; 10: 1060850; Clin Sci (Lond). 2023 Aug; 137(16): 1311–1332). 

Thank you for the above comments regarding appropriate referencing, these have now been added to the text. 

  • Authors should clarify the statistical tests used in their data analysis and the rationale for using those tests. For example, need to expand on PCA. Please describe further for the Kaiser criterion and the varimax method. Why Pearson correlation analysis/factor was necessary? The statistical analysis should be in a separate section (Lines 356-368).

We have added a separate section for the statistical analysis as requested. We had 4 GDM and 5 normal placentae, these were further separated into medium/large and small EV populations. These were then subjected to a Pearson correlation in order to assess the strength of the linear relationships between these populations.  For statistical analysis we used the Perseus software platform, which is specifically designed for proteomics comparison analysis, and their PCA is based on singular value decomposition as described in their paper Tyanova, S., Temu, T., Sinitcyn, P. et al. The Perseus computational platform for comprehensive analysis of (prote)omics data. Nat Methods 13, 731–740 (2016). https://doi.org/10.1038/nmeth.3901. We have now added this to the methods section of our paper. 

  • Figure 2 C should be clear enough to the reader.

This is addressed above, and we have also adapted the text on the figure 2C in the manuscript. 

  • Lines 300-301: How exactly were the participants recruited? What were the inclusion/exclusion criteria?

The participants were approached on the day of delivery and asked to participate. Written consent was taken with ethical approval of Central Oxfordshire Research Ethics Committee C (REFS 07/H0607/74 & 07/H0606/148). All participants were pregnant women scheduled for elective Caesarean section. In order to diagnose GDM, oral glucose tests (OGTT) were performed at 26-28 weeks of gestation. Cut-off values established by guidelines from the Oxford University Hospital for GDM diagnosis were either a fasting plasma glucose level ≥ 5.1 mmol/L and/or a glucose level ≥10.1 mmol/L at one hour and/or ≥ 8.5 mmol/L at two hours after a 75 g oral glucose load. Exclusion criteria for this study led to the omission of women with multiple pregnancies, as well as patients with pre-pregnancy diabetes, cardiovascular disease or pre-eclampsia. Women without abnormal OGTT results formed a control group. We also added this to the text. 

  • Lines 274-282: Authors acknowledge some limitations of the study, but these are unclear and should be further discussed.

We have now expanded on the limitation paragraph.  

  • Lines 287-292: The study comes to a weak conclusion. The conclusions would benefit from giving consideration to how the paragraphs are structured and the thesis of each paragraph.

We fully take on board your comments and have added a separate conclusion paragraph now.  

Round 2

Reviewer 3 Report

Comments and Suggestions for Authors

No further comments.